cognition/psychology

cross-cultural comparison, theory of mind, referential communication, perspective-taking, altercentric interference

**Author for correspondence:**
J. Jessica Wang
e-mail: j.wang40@lancaster.ac.uk

# Perspective-taking across cultures: shared biases in Taiwanese and British adults

J. Jessica Wang[1], Philip Tseng[2], Chi-Hung Juan[3], Steven Frisson[4] and Ian A. Apperly[4]

[1]Department of Psychology, Lancaster University, Lancaster LA1 4YF, UK
[2]Graduate Institute of Humanities in Medicine, Taipei Medical University, Taipei 11031, Taiwan, ROC
[3]Institute of Cognitive Neuroscience, National Central University, Jhongli 32001, Taiwan, ROC
[4]School of Psychology, University of Birmingham, Birmingham B15 2TT, UK

JJW, 0000-0002-4829-5907

The influential hypothesis by Markus & Kitayama (Markus, Kitayama 1991. *Psychol. Rev.* **98**, 224) postulates that individuals from interdependent cultures place others above self in interpersonal contexts. This led to the prediction and finding that individuals from interdependent cultures are less egocentric than those from independent cultures (Wu, Barr, Gann, Keysar 2013. *Front. Hum. Neurosci.* **7**, 1–7; Wu, Keysar. 2007 *Psychol. Sci.* **18**, 600–606). However, variation in egocentrism can only provide indirect evidence for the Markus and Kitayama hypothesis. The current study sought direct evidence by giving British (independent) and Taiwanese (interdependent) participants two perspective-taking tasks on which an other-focused 'altercentric' processing bias might be observed. One task assessed the calculation of simple perspectives; the other assessed the use of others' perspectives in communication. Sixty-two Taiwanese and British adults were tested in their native languages at their home institutions of study. Results revealed similar degrees of both altercentric and egocentric interference between the two cultural groups. This is the first evidence that listeners account for a speaker's limited perspective at the cost of their own performance. Furthermore, the shared biases point towards similarities rather than differences in perspective-taking across cultures.

## 1. Introduction

Theory of mind (ToM) is the ability to represent others' mental states and understand that they could differ from our own mental states. This ability is fundamental to successful communication with others. Uncomplicated mental states of others, such as *what* someone can see (level-1 visual perspective)

is understood from as young as 24 months of age [1] and is calculated with ease by healthy adults [2]. Curiously, when a communicative context demands integration of information about visual perspective with incoming utterances from a communicative partner, adults show high rates of egocentric errors by failing to *use* what they know about their partner's perspective [3,4]. This points towards a dissociation between the calculation of others' visual perspective and using the calculated information appropriately, alongside an egocentric bias to interpret utterances according to one's own perspective.

Studies on the ways in which self and other are conceptualized offer insights into the origin of egocentric bias. Markus & Kitayama [5] argued that concepts of self are shaped by the cultural frameworks in which individuals operate. Most Western cultures emphasize attention to self, which leads to construal of an independent self. By contrast, most East Asian cultures emphasize attention to others, leading to construal of an interdependent self, which is inclusive of others. Furthermore, individuals with an interdependent self-construal would place others' needs and knowledge above one's own in interpersonal contexts. Following the Markus and Kitayama framework, one would expect East Asian communicators to be subject to little or no egocentric interference as others' perspectives are prioritized above their own. By contrast, Western communicators who operate with a focus on themselves would be subject to interference from their own perspective when required to take others' perspectives.

This hypothesis was tested by Wu & Keysar [6] who gave Chinese (East Asian) and American (Western) participants a referential communication task, also known as the director task. In this task, participants were required to follow a director's instructions to rearrange a number of objects on a grid. Critically, the director was 'ignorant' in that she had limited visual access to the grid, as some of the slots on the grid were blocked from her view. Therefore, her seemingly ambiguous instructions (e.g. 'move the block' when there are two blocks in the participants' view) can only be disambiguated by using her perspective to interpret her utterance (i.e. noting that only one of the blocks is in the director's view). Participants' eye movements were recorded as they resolved reference. The American participants clearly attended the additional block in their privileged view by fixating more frequently on this distractor and being slower to make a decisive fixation on the correct block than their Chinese counterparts. This led Wu and Keysar to conclude that the Chinese participants' interdependent self-construal allowed them to focus their attention on the director's perspective, leading them to show a minimal amount of egocentrism.

Wu et al. [7] later carried out a fine-grained time-course analysis on the eye movement data from the Wu & Keysar [6] study. The new analysis revealed that the Chinese participants were as likely to attend to the distractor as their American counterparts immediately after receiving the director's instructions. However, the Chinese participants were considerably quicker than their American counterparts to suppress this initial egocentric tendency and direct their attention to the correct referent in the common ground. This finding brings Wu and Keysar's original interpretation of the Markus & Kitayama [5] proposal into question. If the Chinese participants did prioritize others' perspectives above their own, then Wu et al. should have observed no traces of egocentrism, which was not the case. Additionally, the Wu et al. finding highlights the possibility of shared processes underlying perspective-taking and communication across cultures, i.e. to interpret others' utterances from one's own perspective in the first instance and make corrections when required (see also [8,9], for similar proposals).

Recent evidence also points towards shared basic mechanisms underlying visual perspective-taking among Chinese (East Asian) and British (Western) adults [10]. When making judgements about what another person can see (level-1 visual perspective), both groups of participants traced the person's line of sight to determine whether an object can be seen. When making judgements about the position of an object relative to the person's body (level-2 visual perspective), both groups of participants mentally rotated their own positions to embody the person's position. Therefore, the speed of the responses increased as the angular disparity between the participants and the person increased. This suggests that the ways in which level-1 and level-2 visual perspectives are calculated are highly similar across East Asian and Western cultures.

However, as described earlier, the difficulty communicators face often lies in *using* information about others' perspectives after the information has been calculated. Therefore, to paint a complete picture of the differences and similarities among communicators from different cultures, it is key to examine both the basic mechanisms for perspective-calculation and the online use of others' perspectives. The current study aimed to do so by operationalizing the Markus & Kitayama [5] hypothesis in a novel way. We modified a director task so that it not only captures egocentric tendencies, as reported by Wu & Keysar [6] and Wu et al. [7], but also the tendencies to spontaneously attend others'

perspectives. This would allow us to critically test the Markus & Kitayama hypothesis [5] that individuals from interdependent cultures place others above self in interpersonal contexts. Taiwanese (East Asian) and British (Western) adults were tested on this modified director task. Additionally, we tested participants on a level-1 visual perspective-taking task [2] to verify whether shared basic perspective-taking mechanisms can be detected with a different task to that employed by Kessler et al. [10]. These two tasks allowed us to capture the relative influence of one's own perspective versus others' perspectives on Taiwanese versus British individuals in processing perspectives and using this information to resolve reference.

The Samson et al. [2] task captures two key mechanisms in level-1 visual perspective-taking, namely the interference from one's own perspective on judgements about others' perspectives (egocentric interference) and the interference from others' perspectives on judgements about one's own perspective (altercentric interference). Studies of level-1 visual perspective-taking with various Western adult samples [2,11–14] consistently showed both egocentric interference and altercentric interference. This suggests that, firstly, Western adults' own perspective is a constant source of interference. Secondly, the observation of altercentric interference suggests that Western adults compute others' level-1 visual perspective relatively automatically. Furthermore, this computation is relatively free of cognitive effort [13,14] and is unlikely to be solely stimulus-driven ([12], cf. [15]). Therefore, it is expected that the Taiwanese group will show at least a comparable, if not greater, degree of altercentric interference to that observed in Western adults. This is because the Markus and Kitayama framework suggests that the Taiwanese participants should be more attuned to others' perspectives than the British participants. A second empirical question is whether the Taiwanese group will suffer any egocentric interference when considering others' perspectives on a simple perspective-taking task. The Markus & Kitayama [5] hypothesis would predict little or no egocentrism from the Taiwanese group, given their interdependent self-construal. By contrast, the Wu et al. [7] study suggests that it is at least possible to observe traces of egocentrism in Eastern communicators. The current study aimed to reveal whether the Taiwanese group suffer any egocentric interference, and if so, whether the degree to which they are egocentric differs from the British group.

The director task was modified so that we could capture not only the typically observed egocentric interference, but also a novel altercentric interference. This would allow us to critically test the influential hypothesis that individuals from interdependent cultures place others above self in interpersonal contexts [5]. In additional to the usual 'ignorant' director, we introduced a second, 'informed', director to the director task, who shares the participants' privileged perspective. Both directors were present throughout the task. Therefore, when participants were required to follow the informed director's instructions, there is the possibility of interference from the perspective of the ignorant director, which is also the 'collective perspective' shared between both directors and the participant (figure 1). This additional processing is analogous to the altercentric interference previously described, and it would provide positive evidence of participants attending to the ignorant director's perspective when unnecessary. According to the Markus & Kitayama [5] hypothesis, this altercentric interference should be observed in the Taiwanese group only.

## 2. Method

### 2.1. Participants

Thirty-one participants from the University of Birmingham in the UK and 31 participants from the National Central University in Taiwan were tested in return for a small honorarium. Both groups of participants were recruited through long-standing research participation pools managed by the respective research institutions (a SONA system at the University of Birmingham; a participation Facebook group at the National Central University). The two groups of participants were matched on years of age, gender and the subject and level of study. The sample size and sample-matching criteria were informed by Wu & Keysar [6]. Based on their observed effect size of 0.123 for the critical interaction between experimental versus control condition and Eastern versus Western culture, power analysis conducted in G*Power indicated that a total sample size of 26 would be necessary for 95% power to detect similar effects at $p < 0.05$. Since there were no available data to indicate the size of between-culture differences on the visual perspective-taking task, we sought the largest sample of Taiwanese participants in the available testing time, which substantially exceeded the requirements of the above power analysis and the original sample size of 20 participants from each culture employed by Wu and co-workers. All participants were tested at their home institution. The mean age for both groups was 21.19 years with 16 males (mean age 21.31 years) and 15 females (mean age 21.07 years)

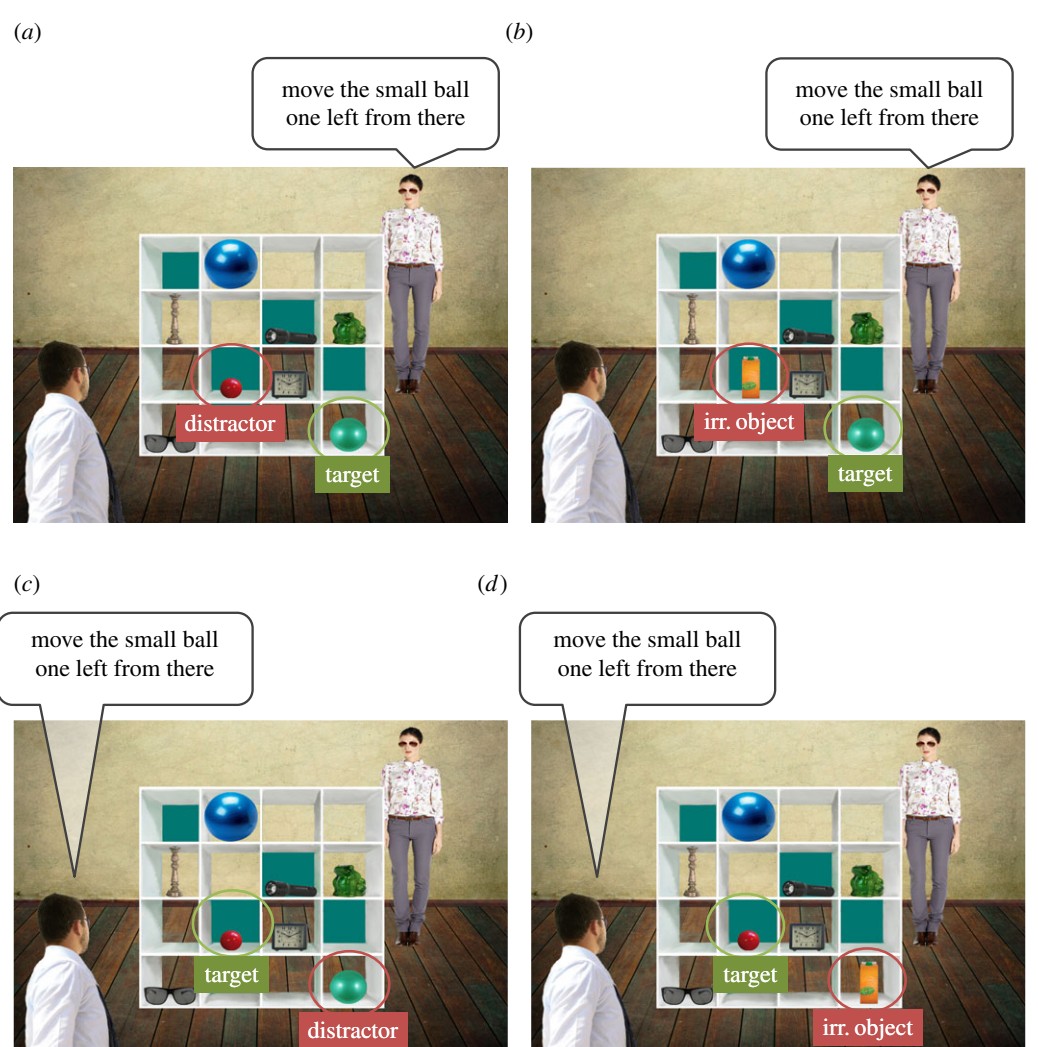

**Figure 1.** Examples of each condition presented to the British participants. The Taiwanese participants were presented with directors of East Asian appearance and Mandarin audio instructions. The ignorant director's perspective coincided with the shared/collective perspective. An example of the (a) ignorant director-experimental condition, (b) ignorant director-control condition, (c) informed director-experimental condition and (d) informed director-control condition. The experimental condition (a) was designed to capture interference from participants' own perspective. Failure to take the ignorant director's perspective and ignore participants' own perspective would lead participants to fixate and/or select the distractor (the red ball in this example) rather than the target (the green ball). The additional processing cost in (a) compared with (b) is termed egocentric interference. The informed director's perspective coincided with participants' privileged perspective. The experimental condition (c) was designed to capture interference from the collective perspective. Failure to take the informed director's perspective and ignore the collective perspective would lead participants to fixate and/or select the distractor (the green ball that is in the shared view) rather than the target (the red ball that is in participants' privileged view). The additional processing cost in (c) compared with (d) is termed altercentric interference.

in each group. Ethical approval has been granted by the Ethics Committee at the University of Birmingham (reference no.: ERN_09-719).

## 2.2. Design and procedure

Each participant completed the adapted computer-based version of the director task [3,4,16,17], which lasted for about 30 min. This was followed by a visual perspective-taking task [2], which lasted for about 20 min. Participants' eye movements were recorded during the director task. Recording of eye movements made it possible to capture effects occurring earlier than overt behavioural responses. It also offered the possibility to evaluate the convergence of effects across a number of measures. Response times and response errors were recorded on both tasks. The Taiwanese participants were instructed and

tested in Mandarin, the British participants were instructed and tested in English. All verbal stimuli were translated into Mandarin and back-translated into English by two English–Mandarin bilingual speakers to check for validity. The images of the directors in the director task and the human avatars in the visual perspective-taking task were of East Asian appearance in the tasks used for the Taiwanese participants, and of Western appearance in the task used for the British participants.

### 2.2.1. The director task

A $2 \times 2 \times 2$ mixed design was constructed with cultural group (British, Taiwanese) as the between-participants factor, and condition (experimental, control) and target perspective (ignorant director, informed director) as the within-participants factors. A total of 32 images were presented, half of which corresponded to the experimental condition, the other half corresponded to the control condition. Each image contained a $4 \times 4$ grid, a female director shown to be standing behind the grid, and a male director shown to be standing in front of the grid, hence sharing the same view of the grid as the participants (figure 1). There were eight different objects on each grid, two to four of which were occluded from the female director's view. Each grid had an additional one to three occluded slots unoccupied by objects. When an image appeared, participants had 5000 ms to examine the image before hearing three to five instructions from the directors, one of which was a critical instruction.

The target perspective was indicated by the directors' voices. The ignorant director was associated with a female voice; the informed director was paired with a male voice. In the experimental condition, the object that best fitted a critical instruction differed between the shared perspective and the privileged perspective (figure 1). Each image in the experimental condition was matched with a highly similar image in the control condition in which the distractor was replaced by an irrelevant object, which did not compete with the target as referent. All other aspects of the images in the matched experimental and control conditions were identical. Each control image was presented at least eight images apart from its matched image in the experimental condition. Participants were equally likely to see an experimental image before its control image as they were to see the reverse order.

A total of 128 instructions were presented auditorily, of which 32 were critical instructions. The remaining 96 instructions were fillers, 32 of which contained scalar adjectives and 16 contained non-scalar adjectives (e.g. blue). All sentences were spliced together from individually recorded words to eliminate the possibility for participants to use co-articulation to identify a referent prior to the onset of the adjective or noun. The critical instructions were 'move the [scalar adjective] [noun] one [directional word] from there' for the British group, '把 [形容詞] [名詞] 向 [方向詞] 移一格' for the Taiwanese group. A complete list of the critical instructions can be found in the electronic supplementary material. The sentences in English and Mandarin were structured so that the onset time of the adjectives and the directional words were matched across the two languages. The interval between the onset of each instruction for a particular item was fixed at 7810 ms to ensure that participants from both cultural groups had the same amount of time to view the grid. Participants were allowed to respond as early as 913 ms from the onset of a sentence. This corresponded to the onset of a critical adjective in critical instructions and fillers containing adjectives or critical noun in fillers that did not contain adjectives. The mean sentence duration was 3810 ms. Participants could respond until 4000 ms after the offset of the sentences. Participants responded with a computer mouse by performing a 'drag and drop' motion as if moving the selected object from one slot to another. Response time was measured from the onset of the adjective to the mouse button press. The mean response time was 3174 ms for the British group and 3152 ms for the Taiwanese group.

We recorded participants' eye movements with an Eyelink 1000 using the tower configuration (SR Research), recording at 1000 Hz. Participants were positioned on a chin rest 60 cm from a 24 inch computer screen. A standard 13-point calibration was carried out before each block. The images subtended 26.93° (width) by 20.15° (height). We drew interest areas around each slot on the grid, which subtended 3.25° (width) by 3.15° (height). We presented eight images per test block, allowing participants to break between the blocks. Four running versions of the experiment were generated by rotating the order of the blocks and by reversing the presentation order of the images. Participants practised on two additional images prior to the four test blocks.

### 2.2.2. The visual perspective-taking task

A $2 \times 2 \times 2$ mixed design was constructed with cultural group (British, Taiwanese) as the between-participants factor, and perspective (self, other) and congruency (congruent, incongruent) as the within-participants factors. The perspective factor refers to the perspective participants were cued to

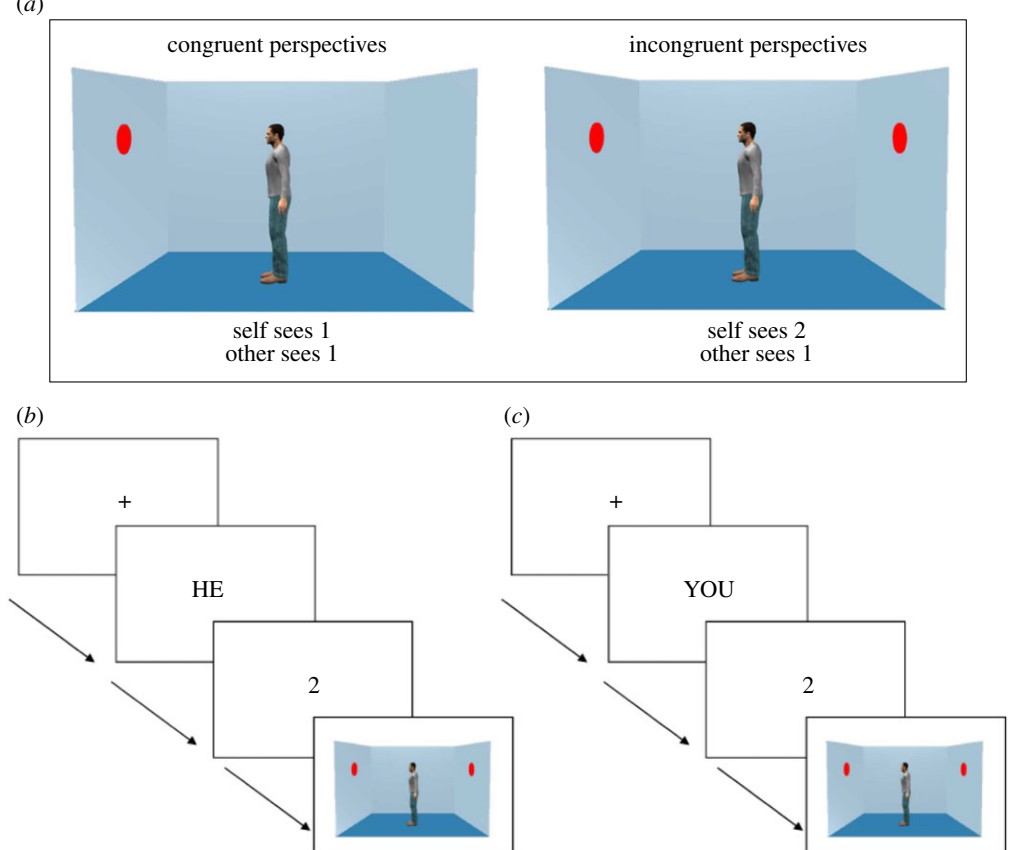

**Figure 2.** Examples of (*a*) test images from the congruent condition and the incongruent condition, (*b*) a trial sequence from the other-incongruent condition and (*c*) the self-incongruent condition. The correct response for (*b*) is 'no' and for (*c*) is 'yes'.

judge on each trial, which was either their own perspective or the avatar's perspective (figure 2). The congruency factor refers to the congruency between the self and other perspectives. In the congruent condition, participants and the avatar shared the same perspective (i.e. they saw the same number of discs), whereas the incongruent condition posed a difference between the two perspectives. The procedure was identical to that of Samson *et al*. [2] with avatars matching the gender and cultural group of the participants, resulting in four versions of the task. The trial sequence is illustrated in figure 2, detailed descriptions can be found in the electronic supplementary material. The mean response time for the British participants was 563 and 494 ms for the Taiwanese participants.

Upon completion of the experiment, participants were asked to explain any strategies they might have employed during the tasks before being given a full debrief. None of the participants mentioned any strategy that would have allowed them to succeed on either task without perspective-taking.

# 3. Results

## 3.1. Director task

Trials containing non-egocentric/altercentric errors (1.65% in the British group, 4.06% in the Taiwanese group), response time-outs (0.08% in the British group, 1.01% in the Taiwanese group) and technical faults (0.08% in the British group, 0.02% in the Taiwanese group) were excluded prior to the analysis. Descriptive statistics can be found in table 1.

Linear mixed effects models were fitted to response times and latency to final target fixation, using the lmer() function from the lme4 package in R [18]. Generalized linear mixed effects models were fitted to egocentric/altercentric errors and proportion of trials containing fixations on distractor, using the glmer() function from the lme4 package in R. The fixed effects for the models for response times, latency to final target fixation and fixation on distractor were cultural group (British, Taiwanese), condition (experimental, control) and target perspective (ignorant director, informed director). Only

**Table 1.** Condition means and standard deviations (s.d.) from (a) behavioural and (b) eye movement data from the director task.

| (a) | | | response time (ms) | | egocentric/altercentric error rate (%) | |
|---|---|---|---|---|---|---|
| culture | perspective | condition | mean | s.d. | mean | s.d. |
| Taiwanese | ignorant dir | control | 3226 | 402 | 0.00 | 0.00 |
| | | experimental | 3304 | 377 | 5.24 | 9.56 |
| | informed dir | control | 2946 | 355 | 0.00 | 0.00 |
| | | experimental | 3012 | 327 | 1.21 | 3.76 |
| British | ignorant dir | control | 3207 | 403 | 0.00 | 0.00 |
| | | experimental | 3269 | 388 | 7.66 | 11.94 |
| | informed dir | control | 3013 | 361 | 0.00 | 0.00 |
| | | experimental | 3049 | 338 | 2.82 | 8.36 |

| (b) | | | latency to final target fixation (ms) | | prop. trials containing fixations on distractor | |
|---|---|---|---|---|---|---|
| culture | perspective | condition | mean | s.d. | mean | s.d. |
| Taiwanese | ignorant dir | control | 2666 | 411 | 0.597 | 0.182 |
| | | experimental | 2729 | 429 | 0.798 | 0.157 |
| | informed dir | control | 2337 | 368 | 0.319 | 0.158 |
| | | experimental | 2412 | 331 | 0.637 | 0.156 |
| British | ignorant dir | control | 2605 | 480 | 0.589 | 0.180 |
| | | experimental | 2661 | 414 | 0.722 | 0.206 |
| | informed dir | control | 2347 | 448 | 0.339 | 0.141 |
| | | experimental | 2454 | 361 | 0.565 | 0.204 |

trials containing correct responses were included in the response times and eye movements analyses. The model for egocentric/altercentric errors only included trials from the experimental condition, as there were no egocentric/altercentric errors in the control condition. Therefore, the model for egocentric/ altercentric errors only included cultural group and target perspective as fixed effects. All fixed effects were included as both main effects and interactions in all models. All fixed effects were coded with contrast coding. Participant and grid image were included as random effects for all models. We attempted to fit models with maximal random effect structure to all models [19]. The terms of the fitted models can be found in the electronic supplementary material. The fitted models were used to determine the statistical significance of a given main effect or interaction. This was achieved by removing one main effect or interaction term from the fitted model at a time, and comparing the models with versus without a given effect (for a summary of the analyses, see table 2).

### 3.1.1. Behavioural data analysis

Analysis on response times revealed a non-significant main effect of cultural group, a significant main effect of condition (experimental > control) and a significant main effect of target perspective (ignorant director > informed director, figure 3). There were no significant interactions. Critically, the interaction effect between cultural group, condition and target perspective was not significant. We calculated Bayesian factor ($BF_{01}$) to quantify evidence for a null model relative to an alternative model. The null model includes all main effects and interaction terms apart from the critical three-way interaction between culture, condition and perspective. The alternative model which includes a three-way interaction term was compared against the null model. The Bayes factor was calculated from the

**Table 2.** Output from model comparisons on (*a*) response time, (*b*) egocentric/altercentric error, (*c*) latency to final target fixation and (*d*) proportion of trials containing fixations on distractor from the director task.

| | $\beta$ | s.e. | $\chi^2$ | d.f. | *p*-value | BF01 |
|---|---|---|---|---|---|---|
| **(*a*) response time** | | | | | | |
| condition | 62.95 | 22.54 | 7.61 | 1 | 0.006 | — |
| culture | 15.59 | 93.22 | 0.03 | 1 | 0.867 | — |
| persp | −252.21 | 90.36 | 6.36 | 1 | 0.012 | — |
| condition × culture | −9.98 | 45.07 | 0.05 | 1 | 0.825 | — |
| condition × persp | 1.83 | 44.39 | <0.01 | 1 | 0.967 | — |
| culture × persp | 88.05 | 97.87 | 0.79 | 1 | 0.374 | — |
| condition × culture × persp | 2.76 | 88.78 | <0.01 | 1 | 0.975 | 41.49 |
| **(*b*) egocentric/altercentric errors** | | | | | | |
| culture | 0.78 | 0.81 | 0.85 | 1 | 0.356 | — |
| persp | −3.85 | 2.39 | 7.41 | 1 | 0.006 | — |
| culture × persp | 0.06 | 1.54 | <0.01 | 1 | 0.969 | 31.15 |
| **(*c*) latency to final target fixation** | | | | | | |
| condition | 60.48 | 36.47 | 2.57 | 1 | 0.109 | — |
| culture | −24.91 | 98.25 | 0.06 | 1 | 0.800 | — |
| persp | −254.14 | 90.76 | 6.44 | 1 | 0.011 | — |
| condition × culture | 13.85 | 69.13 | 0.04 | 1 | 0.842 | — |
| condition × persp | 15.95 | 69.35 | 0.05 | 1 | 0.819 | — |
| culture × persp | 139.79 | 120.72 | 1.09 | 1 | 0.255 | — |
| condition × culture × persp | 65.28 | 130.67 | 0.25 | 1 | 0.618 | 36.76 |
| **(*d*) prop. trials containing fixations on distractor** | | | | | | |
| condition | 1.40 | 0.02 | 20.59 | 1 | <0.001 | — |
| culture | −0.07 | 0.16 | 0.19 | 1 | 0.660 | — |
| persp | −1.32 | 0.38 | 9.23 | 1 | 0.002 | — |
| condition × culture | −0.35 | 0.24 | 2.13 | 1 | 0.145 | — |
| condition × persp | −0.04 | 0.44 | 0.01 | 1 | 0.933 | — |
| culture × persp | 0.07 | 0.28 | 0.07 | 1 | 0.795 | — |
| condition × culture × persp | −0.01 | 0.46 | <0.01 | 1 | 0.993 | 43.56 |

Bayes information criteria (BIC) obtained from the null and alternative models [20]. The $BF_{01}$ suggests that the alternative model was 41.49 times less favourable than the null model. This suggests that a three-way interaction between cultural group, condition and target perspective is highly unlikely.

Analysis on egocentric/altercentric error rates revealed a non-significant main effect of cultural group and a significant main effect of target perspective (ignorant director > informed director). The interaction effect between cultural group and target perspective was not significant. The $BF_{01}$ suggests that the alternative model was 31.15 times less favourable than the null model. This suggests that an interaction between cultural group and target perspective is highly unlikely.

### 3.1.2. Eye movement data analysis

The eye movement data were summarized as the latency to the final target fixation prior to a correct selection, matching Wu & Keysar's [6] report. We also summarized the eye movement data as the proportion of trials containing fixations on the distractor to provide a complementary approach to the analysis based on target items.

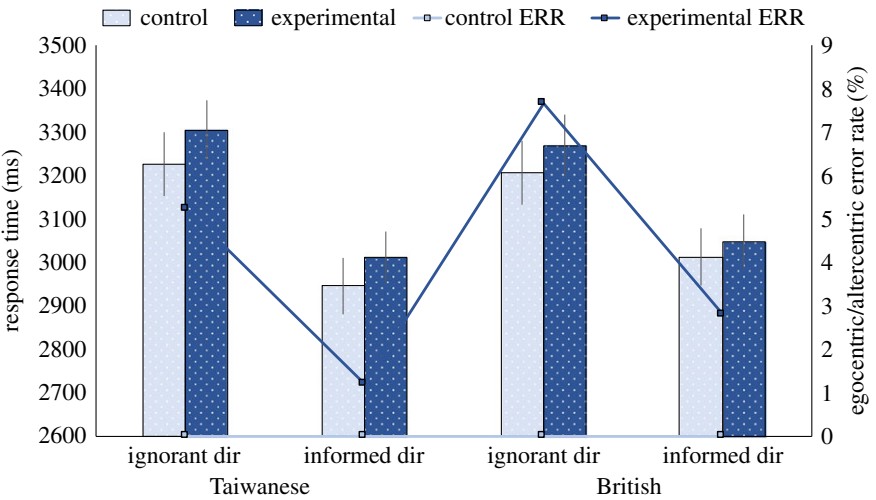

**Figure 3.** Response times and error rates (ERR) from the director task. Error bars correspond to standard errors (s.e.).

Analysis on latencies to final target fixation revealed only a significant main effect of target perspective (ignorant director > informed director). Once again, the interaction effect between cultural group, condition and target perspective was not significant. Bayesian analysis suggests that a model which includes the three-way interaction was 36.76 times less favourable than a null model which includes all other main effect and interaction terms apart from the three-way interaction term. This suggests that a three-way interaction between cultural group, condition and target perspective is highly unlikely.

Analysis on the proportion of trials containing fixations on distractor revealed a significant main effect of condition (experimental > control) and a significant main effect of target perspective (ignorant director > informed director). Critically, the interaction effect between cultural group, condition and target perspective was not significant. Bayesian analysis suggests that a model which includes the three-way interaction was 43.56 times less favourable than a null model which includes all other main effect and interaction terms apart from the three-way interaction term. This suggests that a three-way interaction between cultural group, condition and target perspective is highly unlikely. Furthermore, the eye movement data clearly converged with the behavioural measures in suggesting that the two cultural groups performed very similarly.

## 3.2. Visual perspective-taking task

Trials containing response time-outs (2.71% in the British group, 1.23% in the Taiwanese group) and technical faults (0.02% in the British group, 0% in the Taiwanese group) were excluded prior to the analysis. A British participant failed to perform above chance level in the self condition; therefore, their data along with their matched Taiwanese participant's data were removed prior to the analysis.

A linear mixed effects model was fitted to response times, with a generalized linear mixed effects model fitted to error rates using the lme4 package in R [18]. The fixed effects for the models were cultural group (British, Taiwanese), congruency (congruent, incongruent) and perspective (other, self). Only trials containing correct responses were included in the response times analyses. All fixed effects were included as both main effects and interactions in all models. All fixed effects were coded with contrast coding. Participant and trial image were included as random effects for all models. We attempted to fit models with maximal random effect structure to both response time and error rate models [19]. The terms of the fitted models can be found in the electronic supplementary material. The fitted models were used to determine the statistical significance of a given main effect or interaction (for a summary of the results, see table 3).

Analysis on response times revealed a significant main effect of cultural group (British > Taiwanese), a main effect of congruency (congruent < incongruent) and a significant main effect of perspective (other > self, figure 4). There was a significant interaction effect between congruency and perspective, which was driven by similar response times in the other-congruent and self-congruent condition, $t_{59} = 1.41$, $p = 0.165$, and a slower response to the other-incongruent condition compared with the

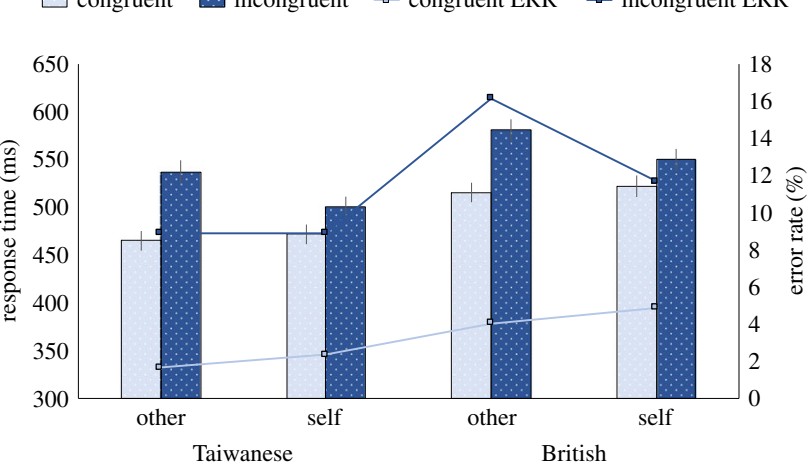

**Figure 4.** Response times and error rates (ERR) from the visual perspective-taking task. Error bars correspond to standard errors (s.e.).

**Table 3.** Outputs from model comparisons on data from the visual perspective-taking task.

| | $\beta$ | s.e. | $\chi^2$ | d.f. | $p$-value | BF01 |
|---|---|---|---|---|---|---|
| **(a) response time** | | | | | | |
| congruency | 49.32 | 7.36 | 22.08 | 1 | <0.001 | — |
| culture | 48.86 | 14.27 | 10.70 | 1 | 0.001 | — |
| persp | −14.39 | 4.21 | 10.75 | 1 | 0.001 | — |
| congruency × culture | −3.52 | 9.43 | 0.13 | 1 | 0.715 | — |
| congruency × persp | −41.83 | 7.31 | 26.08 | 1 | <0.001 | — |
| culture × persp | 5.91 | 9.47 | 0.39 | 1 | 0.535 | — |
| congruency × culture × persp | 12.27 | 17.00 | 0.52 | 1 | 0.473 | 56.50 |
| **(b) errors** | | | | | | |
| congruency | 1.43 | 0.14 | 34.40 | 1 | <0.001 | — |
| culture | 0.66 | 0.18 | 11.96 | 1 | <0.001 | — |
| persp | 0.07 | 0.20 | 0.12 | 1 | 0.725 | — |
| congruency × culture | −0.32 | 0.26 | 1.56 | 1 | 0.212 | — |
| congruency × persp | −0.41 | 0.35 | 1.27 | 1 | 0.259 | — |
| culture × persp | −0.27 | 0.29 | 0.86 | 1 | 0.354 | — |
| congruency × culture × persp | −0.24 | 0.51 | 0.20 | 1 | 0.653 | 68.49 |

self-incongruent condition, $t_{59} = 5.40$, $p < 0.001$, $d = 0.70$. The response times on the incongruent trials were larger than on the congruent trials in both the other and self conditions, $t$'s > 5.72, $p$'s < 0.001, $d > 0.74$. No other interaction effect was significant. The three-way interaction between cultural group, congruency and perspective was not significant. Bayesian analysis suggests that a model which includes the three-way interaction was 56.50 times less favourable than a null model which includes all other main effect and interaction terms apart from the three-way interaction term. This suggests that a three-way interaction between cultural group, congruency and perspective is highly unlikely.

Analysis on error rates revealed a significant main effect of cultural group (British > Taiwanese), a main effect of congruency (congruent < incongruent), with no significant main effect of perspective or any interaction. Critically, three-way interaction between cultural group, congruency and perspective was not significant. Bayesian analysis suggests that a model which includes the three-way interaction was 68.49 times less favourable than a null model which includes all other main effect and interaction terms apart from the three-way interaction term. This suggests that a three-way interaction between cultural group, congruency and perspective is highly unlikely.

## 3.3. Correlation between the director task and the visual perspective-taking task

To determine whether participants' tendencies to suffer egocentric interference and altercentric interference were consistent across the two perspective-taking tasks, scores of egocentric interference and altercentric interference from the two tasks were correlated with each other. For the purpose of conducting the correlational analyses, adjusted scores of egocentric interference and altercentric interference were calculated to account for general differences in processing cost across tasks. The adjusted egocentric interference score from the director task was calculated from the response times and error rate scores using (ignorant director-experimental condition − ignorant director-control condition) ÷ (ignorant director-experimental condition + ignorant director-control condition). The adjusted altercentric interference score from the director task was calculated using informed director-experimental and informed director-control conditions. The adjusted egocentric interference score from the visual perspective-taking task was calculated from the response times and error rate scores in (other-incongruent condition − other-congruent condition) ÷ (other-incongruent condition + other-congruent condition), respectively. The adjusted altercentric interference score from the visual perspective-taking task was calculated using self-incongruent and self-congruent conditions. There was no significant correlation between the degree of egocentric interference from the visual perspective-taking task and the director task in response times $r = 0.179$, $p = 0.170$ ($BF_{01} = 2.48$), or error rate $r = -0.120$, $p = 0.360$ ($BF_{01} = 4.13$), and no significant correlation between the degree of altercentric interference from the two tasks in response times, $r = -0.087$, $p = 0.507$ ($BF_{01} = 5.01$) or error rate $r = -0.068$, $p = 0.606$ ($BF_{01} = 5.45$).

# 4. Discussion

To date, cross-cultural studies of perspective-taking have narrowly focused on the degrees to which individuals from independent versus interdependent cultures are egocentric [6,7]. This provides little positive evidence for the hypothesis that individuals from interdependent cultures place others above self in interpersonal contexts [5]. The current study critically examined this hypothesis by including two perspective-taking tasks, one to assess cross-cultural differences in the calculation of simple perspectives and the other to assess the use of others' perspectives in communication. Both tasks afforded the opportunity to capture the typical egocentric interference and a novel altercentric interference, which is key for assessing the degrees of other-focused processing. Results revealed remarkable cross-cultural similarities; both the Taiwanese group (interdependent) and the British group (independent) displayed similar degrees of egocentrism on both tasks. Furthermore, both groups of adults also displayed similar degrees of altercentric interference on both tasks. These findings critically inform approaches to cross-cultural studies in a number of ways, which we discuss below.

## 4.1. Level-1 visual perspective-taking

The Samson et al. [2] visual perspective-taking task was included to capture egocentric interference and altercentric interference when considering simple perspectives. The current study demonstrated for the first time that individuals from an interdependent culture suffer as much egocentric interference as individuals from an independent culture.[1] This suggests that the early egocentric tendency observed by Wu et al. [7] may not always be corrected in time and can have an effect on behavioural responses. Another novel observation was that the Taiwanese participants suffered altercentric interference to a similar degree to the British participants. This suggests that both groups of participants were influenced by others' perspectives to a similar degree. There is ongoing debate about whether altercentric interference observed in level-1 visual perspective-taking (e.g. [15]) arises from processing of the other's perspective, or solely from processing of the other's spatial properties. These alternatives cannot be distinguished within the current study. Nevertheless, the key finding here is that individuals from self-focused culture and other-focused culture showed highly similar degrees of altercentric interference, suggesting that they did not give any special priority to the spatial properties of the avatar or to her perspective. This finding challenges the notion that individuals from interdependent cultural frameworks have a greater focus on others than those from independent cultural frameworks.

---

[1]The main effects of cultural group in response times and error rates observed in this task indicated that the British participants were generally slower and more error prone than the Taiwanese participants. However, the effect of cultural group did not interact with effects of congruency or perspective. This indicates that the degrees of egocentric interference and altercentric inference observed did not vary according to cultural groups.

The observation that both types of interference are equivalent across cultures suggests that the basic mechanisms underlying perspective-taking are likely to be shared. Although the overall processing cost incurred was smaller for the Taiwanese participants than the British participants, the degrees of egocentric interference and altercentric interference were proportional to the processing costs. This is consistent with Kessler *et al.*'s [10] suggestion of shared basic processes involved in calculating visual perspectives. Interestingly, evidence from a developmental study of level-1 visual perspective-taking suggests that the amount of egocentric interference and altercentric interference do not undergo detectable changes with age, even when the overall processing costs reduce with age [21]. Taken together with the current finding, this suggests that the basic mechanisms for level-1 visual perspective-taking are likely to be in place from a young age and are relatively independent of even quite diverse cultural experiences.

## 4.2. Using information about others' perspectives in communication

Where the visual perspective-taking task revealed effects of perspective-calculation, the director task assessed the online use of the perspective information calculated to resolve reference. The modified director task we employed allowed us to capture both egocentric and altercentric interference while participants took the two directors' perspectives in referential communication. As expected, the British participants were influenced by their own privileged perspective when they followed the ignorant director's instructions. Interestingly, the Taiwanese participants were as egocentric as their British counterparts. This finding is consistent with the results from the level-1 visual perspective-taking task. Once again, the Taiwanese participants carried interference from their egocentric perspective to the point of response. This finding may appear to be incompatible with that of Wu and co-workers [6,7], in that they did not find any egocentrism in behavioural responses. However, a likely explanation is that in Wu and co-workers' study, the director delivered ambiguous reference, which cannot be resolved unless her perspective was taken into account. This design probably prompted their participants to at least seek a solution to disambiguate the director's reference. By contrast, in the current study, the referential expressions from the ignorant director most closely matched the distractors in the participants' privileged view. Therefore, rather than being prompted to use the director's perspective, our participants had to actively bear in mind that the ignorant director had limited access of the referential grid in order to make correct responses. Evidence suggests that trials with ambiguous expressions generate considerably lower rates of egocentric errors compared with those with relative expressions within the same task [3]. This difference in design could also explain the patterns of eye movements observed in Wu and co-workers' study and the current study. It seems at least possible that the Chinese participants in Wu and co-workers' study were simply quicker to respond to the prompts offered by the director's ambiguous expression than their American counterparts. In the absence of such prompts, the Taiwanese participants in the current study were influenced by their own perspective as much as their British counterparts. Such variation in performance is likely to correspond to the behaviours observed in varied social contexts. For example, communicators from Hong Kong committed higher rates of egocentric errors on the director task after being primed with Western symbols compared with Chinese symbols or neutral primes [22]. Taken together with the current study, this highlights the variability in the degrees of egocentrism within a cultural group. The variation critically depends on the parameters of the task [3], the social context constructed [22] and even further factors, such as linguistic demand [17], social functioning [23] and motivation [24]. The correspondence between degrees of egocentrism and these factors suggests that the director task probably captures at least some of the characteristics and demands of perspective-taking and communication in real life.[2]

## 4.3. Spontaneous processing of the collective perspective

A surprising finding from the current study is that participants from both cultural groups suffered similar amounts of interference from the collective perspective. The altercentric interference was captured on trials where participants followed instructions from the informed director, who shared

---

[2]Here, we have focused our discussion on a typically developed adult population. Evidence from clinical and developmental populations further supports the ecological validity of the director task. For example, adults with depressive symptoms and children with ADHD showed impaired perspective-taking performances in similar communication tasks [25,26]. Interestingly, adults with autism were shown to perform no differently to typically developed adults in a variation of the director task [27,28]; however, autistic traits in typically developed adults did lead to compromised performance [23].

participants' privileged view of the grid. The observation of altercentric interference during reference resolution suggests that both groups of participants were at least aware of an alternative interpretation of the instructions. This finding is in tension with the Markus & Kitayama [5] hypothesis, which postulates individualists to be confined to their own perspectives. According to this hypothesis, the present experiment should only have observed Taiwanese participants suffering interference from the collective perspective, driven by adaptation of a collective perspective.

What is more, even though there is no reason to suppose that participants must calculate the director's perspective afresh on every trial, in order to suffer altercentric interference in referential communication participants must at some point have calculated both directors' level-1 visual perspectives, and also engaged effortful processes to integrate information about visual perspective with incoming instructions [8]. It is not clear whether such effects would occur in all circumstances. It is possible that the demand to switch between following instructions from the two directors with different perspectives drew attention to both directors. Similar effects have been observed in studies of level-2 visual perspective-calculation, which is considered to be an effortful process (e.g. [29,30]). It has been shown that the task context of mixed versus blocked presentation of self and other trials determined whether (Western) adults spontaneously calculate others' level-2 visual perspectives [31]. This suggests that effortful calculation of others' perspectives could be encouraged when the task context provided participants with reasons to do so. Conversely, if participants were only ever required to use their own perspective to resolve reference, we might observe reduced altercentric interference, and the amount of reduction may differ across cultures. However, such scenarios do not correspond to the demand communication often poses in real life. Instead, communicators frequently face the demand to keep track of multiple communicative partners' perspectives. It is having the possibility to spontaneously keep track of others' perspectives in communication that provides the foundation for timely social interactions in both independent and interdependent cultures.

## 4.4. Perspective-calculation and perspective-use

The level-1 visual perspective-taking task [2] and the modified director task [4] employed in the current study both pointed towards remarkable similarities between the Taiwanese group and the British group. Both cultural groups were influenced by their own perspective when making simple level-1 visual perspective judgements and when using others' perspective in referential communications. Both cultural groups also showed spontaneous calculation of others' perspectives in both contexts. Interestingly, there was no significant correlation between the degrees of egocentric interference or altercentric interference on the two tasks. This finding supports the notion that perspective-calculation and the use of the calculated information are distinct processes [32]. Although effects of egocentric and altercentric interference were observed on both tasks, they probably reflect the respective demands and features occurring at an early calculation stage versus a later use stage of perspective-taking. The finding also suggests that egocentric and altercentric tendencies, although consistently observed across different contexts, are unlikely to operate in a trait-like fashion. Instead, they are likely to incur due to situational processing demands, such as having to make use of both one's own and others' perspectives within a task. Therefore, given appropriate contexts, individuals from interdependent cultures could be egocentric, and for those from independent cultures could be altercentric, as demonstrated in the current study.

## 4.5. Theory of mind in independent and interdependent cultural frameworks

The current study offered a novel way to operationalize the influential Markus & Kitayama [5] hypothesis. We modified a director task so that it not only captures egocentric tendencies, but also the tendencies to spontaneously attend others' perspectives. Remarkable similarities in perspective-taking across Taiwanese and British adults were seen. Such similarities were consistently found on two different perspective-taking tasks. The current study is not the first to find similarities in ToM abilities across cultures that were thought to be independent versus interdependent. Preschoolers from Chinese versus American cultures were shown to have an emerging ability to distinguish appearance and reality around the same age [33]. Furthermore, despite Chinese children's superior executive functioning, they still showed similar performances to their American counterparts on false belief tasks [34]. Such cross-cultural similarities were found even when adults from Chinese and Western cultures engage in belief attribution [35]. Along with these findings, the current study critically challenges the dichotomist view of cross-cultural differences. It is clear that the cross-cultural

frameworks by Markus & Kitayama [5] and others are useful in capturing and describing cross-cultural differences observed on a structural level in cultures and societies. However, it is clear that we cannot take for granted that similar differences will occur on functional and operational levels. In the case of ToM, it is critical to incorporate contextual factors in the model.

Ethics. Ethical approval has been granted by the Ethics Committee at the University of Birmingham (reference no.: ERN_09-719). Informed consent was obtained from participants.

Data accessibility. Trial level data and the R code used in the analyses from the current study can be found in the Dryad Digital Repository: https://dx.doi.org/10.5061/dryad.3t2v0f9 [36]. The data are also available within the electronic supplementary material.

Authors' contributions. J.J.W. conceived and designed the study, collected the data, carried out data analysis and drafted the manuscript. P.T. and C.-H.J. assisted with experimental set-up and recruitment at the Taiwanese site, and critically revised the manuscript. S.F. and I.A.A. conceived and designed the study, provided critical discussions about the analysis and critically revised the manuscript. All authors gave final approval for publication.

Competing interests. The authors declare no competing interests.

Funding. This work was supported by a research grant funded by the Economic & Social Research Council to I.A.A. and S.F. (reference no.: ES/J012238/1).

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
