## [Reviewer comments · Royal Society Open Science]

Review History

RSOS-190540.R0 (Original submission)

Review form: Reviewer 1

Is the manuscript scientifically sound in its present form?

Yes

Are the interpretations and conclusions justified by the results?

Yes

Is the language acceptable?

Yes

Is it clear how to access all supporting data?

Yes

Do you have any ethical concerns with this paper?

No

Have you any concerns about statistical analyses in this paper?

No

Recommendation?

Accept with minor revision (please list in comments)

Comments to the Author(s)

This manuscript reports the results of a behavioural and eye-tracking study in two groups of adult participants, one British and one Taiwanese (N=62 in total). The aim was to follow up on a framework and previous evidence suggesting that cultures with independent vs. interdependent self-constructs differ in how much they take into account other people's perspective vs. have an egocentric bias. Participants completed two tasks designed to assess influences of one's and others' perspective on decisions about others' and one's perspective respectively. The results show that there was no evidence that British and Taiwanese groups differed in how much they were influenced by the perspective of others or in their egocentric bias.

This paper is an interesting study that is providing important evidence regarding cross-cultural comparisons of social cognition task performance. The two tasks are appropriate for the aim of the study and the analyses thorough.

My main comment regards some of the terminology used to describe the different task conditions of the Director task. The authors talk about "collective perspective" and "shared perspective" for the condition where the speaker is the director behind the shelves. But to me this is in fact a condition where the perspective is not shared as it differs for the director and the participant. I understand the authors mean the participant has to figure out what the shared perspective is but I think it's misleading. Similarly, the term "collective" would suggest that all the individuals (two directors + participants) have the same perspective, which is not the case. I would suggest maybe calling this the "different perspective" condition, vs. "privileged perspective" or "shared perspective" for the one where the director is on the same side as the participant.

Minor comments

The sentence on p.7 lines 24-29 is unclear, who is "the communicator" and saying "from different cultures" suggest the study looks at interaction between e.g. a British participant and a Taiwanese avatar. I would suggest rephrasing.

Figure 1 indicates that the director's instructions were of the form "Move the small ball one left". This seems to be grammatically incorrect, why wasn't the instructions of the form "Move the small ball one slot left" or "towards the left"? Were there any cases where the move was further than one slot?

The sentence on p.10 lines 33-38 suggests the mean age for the two groups were exactly the same, to the second decimal - was it really the case? How was this perfect match achieved?

The sentence on p. 11 lines 52-50 seems strangely phrased, why use "as" in "as her perspective represented the shared perspective"?

p.13 I am not sure what the sentence "The mean sentence duration was 3810ms, giving participants approximately 4000ms to respond." Does it mean that participants were asked to respond before the sentence was finished? What was the actual responding time limit?

Figure 2: I would suggest giving examples of congruent trials.

p.17 line 17: “There were no other significant interactions.” > “There was no significant interaction.” (you have only talked about main effects prior to this sentence)

There seems to be some duplication of data presented in figures and tables (e.g. Table 3 and Figure 4), I am not sure this is necessary.

Review form: Reviewer 2

Is the manuscript scientifically sound in its present form?

Yes

Are the interpretations and conclusions justified by the results?

Yes

Is the language acceptable?

Yes

Is it clear how to access all supporting data?

Yes

Do you have any ethical concerns with this paper?

No

Have you any concerns about statistical analyses in this paper?

No

Recommendation?

Major revision is needed (please make suggestions in comments)

Comments to the Author(s)

Review RSOS-190540

The manuscript describes a solid experiment, preregistered, on an old hypothesis re cultural differences in perspective taking. The introduction presents a very clear overview of relevant literature, explaining key concepts to a wide audience. However, a perspective on the validity of these measures would strengthen the rationale. The addition of the altercentric condition to the director task is a great advantage, but the issue of validity remains. The study could be presented more briefly, but adds to our understanding of cultural similarities.

Main issues

1. What is the ecological validity of the director task? What are associations with other indices (e.g. proxy reports) of social perspective taking? E.g., in clinical groups (like autism) the perspective task has had inconsistent results.
2. The authors emphasize the focus on the use of perspective but do not provide any account of the empirical literature on cultural differences in this domain.
3. What is the relative importance of eye movement/RT versus behavioral outcomes, would behavioral outcomes not be more meaningful? Is there a rationale for focusing on eye movement other than methodological (sensitivity?)
4. Both tasks aim to measure perspective taking, but show very weak correlation. What does this say about their validity? Would it not have been better to include additional measures to provide information on the cultural

Minor issues.

1. the paper could be shorter, in particular the intro and methods/results. Some methods sections could be moved to appendix
2. The info on participants is minimal, what were recruitment procedures, were any analyses ran to justify the level of cultural identification among the participants?
3. the main effect of culture on error rate is not emphasized in the discussion. How can this contrast be explained given the overall pattern of results?
4. The discussion could be more condensed, several points are repeated, and the studies could be put in a wider theoretical and methodological framework.

Decision letter (RSOS-190540.R0)

03-Jun-2019

Dear Dr Wang,

The editors assigned to your paper ("Perspective-taking Across Cultures: Shared biases in Taiwanese and British Adults") have now received comments from reviewers. We would like you to revise your paper in accordance with the referee and Associate Editor suggestions which can be found below (not including confidential reports to the Editor). Please note this decision does not guarantee eventual acceptance.

Please submit a copy of your revised paper before 26-Jun-2019. Please note that the revision deadline will expire at 00.00am on this date. If we do not hear from you within this time then it will be assumed that the paper has been withdrawn. In exceptional circumstances, extensions may be possible if agreed with the Editorial Office in advance. We do not allow multiple rounds of revision so we urge you to make every effort to fully address all of the comments at this stage. If deemed necessary by the Editors, your manuscript will be sent back to one or more of the original reviewers for assessment. If the original reviewers are not available, we may invite new reviewers.

If your study uses humans or animals please include details of the ethical approval received, including the name of the committee that granted approval. For human studies please also detail

whether informed consent was obtained. For field studies on animals please include details of all permissions, licences and/or approvals granted to carry out the fieldwork.

- Data accessibility

If you wish to submit your supporting data or code to Dryad (<http://datadryad.org/>), or modify your current submission to dryad, please use the following link:
<http://datadryad.org/submit?journalID=RSOS&manu=RSOS-190540>

- Competing interests

- Authors' contributions

- Acknowledgements

- Funding statement

Kind regards,
Alice Power
Editorial Coordinator

on behalf of Dr Antonia Hamilton (Associate Editor) and Essi Viding (Subject Editor)
 openscience@royalsociety.org

Associate Editor's comments (Dr Antonia Hamilton):

The reviewers comments are helpful and I hope that the authors can respond to all of them, with some edits to the intro & discussion as needed.

Comments to Author:

Reviewers' Comments to Author:

Reviewer: 1

Comments to the Author(s)

This manuscript reports the results of a behavioural and eye-tracking study in two groups of adult participants, one British and one Taiwanese (N=62 in total). The aim was to follow up on a framework and previous evidence suggesting that cultures with independent vs. interdependent self-constructs differ in how much they take into account other people's perspective vs. have an egocentric bias. Participants completed two tasks designed to assess influences of one's and others' perspective on decisions about others' and one's perspective respectively. The results show that there was no evidence that British and Taiwanese groups differed in how much they were influenced by the perspective of others or in their egocentric bias.

This paper is an interesting study that is providing important evidence regarding cross-cultural comparisons of social cognition task performance. The two tasks are appropriate for the aim of the study and the analyses thorough.

My main comment regards some of the terminology used to describe the different task conditions of the Director task. The authors talk about "collective perspective" and "shared perspective" for the condition where the speaker is the director behind the shelves. But to me this is in fact a condition where the perspective is not shared as it differs for the director and the participant. I understand the authors mean the participant has to figure out what the shared perspective is but I think it's misleading. Similarly, the term "collective" would suggest that all the individuals (two directors + participants) have the same perspective, which is not the case. I would suggest maybe calling this the "different perspective" condition, vs. "privileged perspective" or "shared perspective" for the one where the director is on the same side as the participant.

Minor comments

The sentence on p.7 lines 24-29 is unclear, who is "the communicator" and saying "from different cultures" suggest the study looks at interaction between e.g. a British participant and a Taiwanese avatar. I would suggest rephrasing.

Figure 1 indicates that the director's instructions were of the form "Move the small ball one left". This seems to be grammatically incorrect, why wasn't the instructions of the form "Move the small ball one slot left" or "towards the left"? Were there any cases where the move was further than one slot?

The sentence on p.10 lines 33-38 suggests the mean age for the two groups were exactly the same, to the second decimal – was it really the case? How was this perfect match achieved?

The sentence on p. 11 lines 52-50 seems strangely phrased, why use “as” in “as her perspective represented the shared perspective”?

p.13 I am not sure what the sentence “The mean sentence duration was 3810ms, giving participants approximately 4000ms to respond.” Does it mean that participants were asked to respond before the sentence was finished? What was the actual responding time limit?

Figure 2: I would suggest giving examples of congruent trials.

p.17 line 17: “There were no other significant interactions.” > “There was no significant interaction.” (you have only talked about main effects prior to this sentence)

There seems to be some duplication of data presented in figures and tables (e.g. Table 3 and Figure 4), I am not sure this is necessary.

Reviewer: 2

Comments to the Author(s)

Review RSOS-190540

The manuscript describes a solid experiment, preregistered, on an old hypothesis re cultural differences in perspective taking. The introduction presents a very clear overview of relevant literature, explaining key concepts to a wide audience. However, a perspective on the validity of these measures would strengthen the rationale. The addition of the altercentric condition to the director task is a great advantage, but the issue of validity remains. The study could be presented more briefly, but adds to our understanding of cultural similarities.

Main issues

1. What is the ecological validity of the director task? What are associations with other indices (e.g. proxy reports) of social perspective taking? E.g., in clinical groups (like autism) the perspective task has had inconsistent results.
2. The authors emphasize the focus on the use of perspective but do not provide any account of the empirical literature on cultural differences in this domain.
3. What is the relative importance of eye movement/RT versus behavioral outcomes, would behavioral outcomes not be more meaningful? Is there a rationale for focusing on eye movement other than methodological (sensitivity?)
4. Both tasks aim to measure perspective taking, but show very weak correlation. What does this say about their validity? Would it not have been better to include additional measures to provide information on the cultural

Minor issues.

1. the paper could be shorter, in particular the intro and methods/results. Some methods sections could be moved to appendix
2. The info on participants is minimal, what were recruitment procedures, were any analyses run to justify the level of cultural identification among the participants?
3. the main effect of culture on error rate is not emphasized in the discussion. How can this contrast be explained given the overall pattern of results?
4. The discussion could be more condensed, several points are repeated, and the studies could be put in a wider theoretical and methodological framework.

Author's Response to Decision Letter for (RSOS-190540.R0)

See Appendix A.

RSOS-190540.R1 (Revision)

Review form: Reviewer 1

Is the manuscript scientifically sound in its present form?

Yes

Are the interpretations and conclusions justified by the results?

Yes

Is the language acceptable?

Yes

Do you have any ethical concerns with this paper?

No

Have you any concerns about statistical analyses in this paper?

No

Recommendation?

Accept with minor revision (please list in comments)

Comments to the Author(s)

The authors have adequately addressed my comments.

I only have one point remaining regarding the response timings, I now understand what was done but I think the sentence is still unclear, I would suggest writing:

“The mean sentence duration was 3810ms. Participants could respond until 4000ms after the onset of the sentences.”

Review form: Reviewer 3 (Dale Barr)

Is the manuscript scientifically sound in its present form?

Yes

Are the interpretations and conclusions justified by the results?

Yes

Is the language acceptable?

Yes

Do you have any ethical concerns with this paper?

No

Have you any concerns about statistical analyses in this paper?

No

Recommendation?

Accept as is

Comments to the Author(s)

This study by Wang et al. presents an interesting comparison of perspective-taking across two cultures (Taiwanese and British) and documents patterns of interference that seem largely similar, suggesting similar underlying mechanisms. Thirty-one participants from each of the two cultural groups completed the 'director' task as well as a visual perspective-taking task. On both tasks, similar results were found across the groups, in contrast with previous work suggesting striking cultural differences (Wu & Keysar, 2007).

I reviewed this manuscript previously for a different journal. My review is not among the previous reviews mentioned in the authors' "Response to Reviews". Still, I was pleased to note that the authors satisfactorily addressed all major concerns I raised in that review, which were (1) that the Samson et al. task may be measuring other things than 'altercentric' interference (now conceded on p. 25 of the manuscript); (2) the lack of by-item analyses (now addressed through the use of linear mixed effects models); and (3) unavailability of data and code (now made available on the Data Dryad repository). I was able to load in the data and reproduce some of the values in Table 2, which gives me confidence the analyses were performed correctly.

I have no issues with the manuscript and congratulate the authors on a very important contribution to the discussion about cultural differences in perspective taking, and look forward to seeing this in the literature.

Signed,
-Dale Barr

Decision letter (RSOS-190540.R1)

23-Sep-2019

Dear Dr Wang:

On behalf of the Editors, I am pleased to inform you that your Manuscript RSOS-190540.R1 entitled "Perspective-taking Across Cultures: Shared biases in Taiwanese and British Adults" has been accepted for publication in Royal Society Open Science subject to minor revision in accordance with the referee suggestions. Please find the referees' comments at the end of this email.

The reviewers and Subject Editor have recommended publication, but also suggest some minor revisions to your manuscript. Therefore, I invite you to respond to the comments and revise your manuscript.

- Ethics statement

- Data accessibility

<http://datadryad.org/submit?journalID=RSOS&manu=RSOS-190540.R1>

- Competing interests

- Authors' contributions

- Acknowledgements

- Funding statement

Please note that we cannot publish your manuscript without these end statements included. We have included a screenshot example of the end statements for reference. If you feel that a given

heading is not relevant to your paper, please nevertheless include the heading and explicitly state that it is not relevant to your work.

Because the schedule for publication is very tight, it is a condition of publication that you submit the revised version of your manuscript before 02-Oct-2019. Please note that the revision deadline will expire at 00.00am on this date. If you do not think you will be able to meet this date please let me know immediately.

on behalf of Dr Antonia Hamilton (Associate Editor) and Essi Viding (Subject Editor)
openscience@royalsociety.org

Associate Editor Comments to Author (Dr Antonia Hamilton):

Thank you for the submission - the reviewers are largely content to recommend publication. One of the referees has a minor suggestion regarding phrasing of a sentence - this can be tweaked during typesetting/proofing by you.

Reviewer comments to Author:

Reviewer: 1

Comments to the Author(s)

The authors have adequately addressed my comments.

I only have one point remaining regarding the response timings, I now understand what was done but I think the sentence is still unclear, I would suggest writing:
"The mean sentence duration was 3810ms. Participants could respond until 4000ms after the onset of the sentences."

Reviewer: 3

Comments to the Author(s)

This study by Wang et al. presents an interesting comparison of perspective-taking across two cultures (Taiwanese and British) and documents patterns of interference that seem largely similar, suggesting similar underlying mechanisms. Thirty-one participants from each of the two cultural groups completed the 'director' task as well as a visual perspective-taking task. On both tasks, similar results were found across the groups, in contrast with previous work suggesting striking cultural differences (Wu & Keysar, 2007).

I reviewed this manuscript previously for a different journal. My review is not among the previous reviews mentioned in the authors' "Response to Reviews". Still, I was pleased to note that the authors satisfactorily addressed all major concerns I raised in that review, which were (1) that the Samson et al. task may be measuring other things than 'altercentric' interference (now conceded on p. 25 of the manuscript); (2) the lack of by-item analyses (now addressed through the use of linear mixed effects models); and (3) unavailability of data and code (now made available on the Data Dryad repository). I was able to load in the data and reproduce some of the values in Table 2, which gives me confidence the analyses were performed correctly.

I have no issues with the manuscript and congratulate the authors on a very important contribution to the discussion about cultural differences in perspective taking, and look forward to seeing this in the literature.

Signed,

-Dale Barr

Author's Response to Decision Letter for (RSOS-190540.R1)

See Appendix B.

Decision letter (RSOS-190540.R2)

01-Oct-2019

Dear Dr Wang,

I am pleased to inform you that your manuscript entitled "Perspective-taking Across Cultures: Shared biases in Taiwanese and British Adults" is now accepted for publication in Royal Society Open Science.

Kind regards,

Lianne Parkhouse
Royal Society Open Science
openscience@royalsociety.org

on behalf of Dr Antonia Hamilton (Associate Editor) and Essi Viding (Subject Editor)
openscience@royalsociety.org

Appendix A

Response to reviewers

Please find point by point response to each of the comments below. Changes are highlighted in yellow in the main manuscript.

Reviewer: 1

Comments to the Author(s)

This manuscript reports the results of a behavioural and eye-tracking study in two groups of adult participants, one British and one Taiwanese (N=62 in total). The aim was to follow up on a framework and previous evidence suggesting that cultures with independent vs. interdependent self-constructs differ in how much they take into account other people's perspective vs. have an egocentric bias. Participants completed two tasks designed to assess influences of one's and others' perspective on decisions about others' and one's perspective respectively. The results show that there were no evidence that British and Taiwanese groups differed in how much they were influenced by the perspective of others or in their egocentric bias.

This paper is an interesting study that is providing important evidence regarding cross-cultural comparisons of social cognition task performance. The two tasks are appropriate for the aim of the study and the analyses thorough.

My main comment regards some of the terminology used to describe the different task conditions of the Director task. The authors talk about "collective perspective" and "shared perspective" for the condition where the speaker is the director behind the shelves. But to me this is in fact a condition where the perspective is not shared as it differs for the director and the participant. I understand the authors mean the participant has to figure out what the shared perspective is but I think it's misleading. Similarly, the term "collective" would suggest that all the individuals (two directors + participants) have the same perspective, which is not the case. I would suggest maybe calling this the "different perspective" condition, vs. "privileged perspective" or "shared perspective" for the one where the director is on the same side as the participant.

RESPONSE: We have thought long and hard about the condition labelling. However we feel that the label 'different perspective' may generate confusion with the experimental/control condition, as these conditions provides a direct contrast of different/same perspective in terms of resolving critical reference. We have modified the target perspective condition labels to "ignorant director" and "informed director", we feel that these labels can be easily associated with the task participants undertook.

Minor comments

The sentence on p.7 lines 24-29 is unclear, who is "the communicator" and saying "from different cultures" suggest the study looks at interaction between e.g. a British participant and a Taiwanese avatar. I would suggest rephrasing.

RESPONSE: this has now been rephrased.

Figure 1 indicates that the director's instructions were of the form "Move the small ball one left". This seems to be grammatically incorrect, why wasn't the instructions of the form "Move the small ball one slot left" or "towards the left"? Were there any cases where the move was further than one slot?

RESPONSE: the sentence was structured in such a way so that the onset time of the critical adjective, the directional word, and the sentence length are matched across English and Mandarin (as explained in the final paragraph on page 12). The full sentence was "move the small ball one left from there", and we have taken the opportunity to make this correction in Figure 1 (the full sentence list included in the appendix was correct).

The sentence on p.10 lines 33-38 suggests the mean age for the two groups were exactly the same, to the second decimal – was it really the case? How was this perfect match achieved?

RESPONSE: The two cultural groups were matched on years of age, therefore the group means were identical. We have now clarified the relevant wording on page 10.

The sentence on p. 11 lines 52-50 seems strangely phrased, why use "as" in "as her perspective represented the shared perspective"?

RESPONSE: this section has now been moved to the end of page 8, where we explain the function of the ignorant director and the informed director.

p.13 I am not sure what the sentence “The mean sentence duration was 3810ms, giving participants approximately 4000ms to respond.” Does it mean that participants were asked to respond before the sentence was finished? What was the actual responding time limit?

RESPONSE: Participants were not specifically instructed to respond before or after the full sentence had ended. The earliest point participants could respond was from the onset of the critical adjective, and the latest point was 4000 after the full sentence have been delivered. We have clarified this point on pages 12-13.

Figure 2: I would suggest giving examples of congruent trials.

RESPONSE: this has been added to page 14.

p.17 line 17: “There were no other significant interactions.” > “There was no significant interaction.” (you have only talked about main effects prior to this sentence)

RESPONSE: this has been corrected (now on page 16).

There seems to be some duplication of data presented in figures and tables (e.g. Table 3 and Figure 4), I am not sure this is necessary.

RESPONSE: Table 3 has now been removed.

Reviewer: 2

The manuscript describes a solid experiment, preregistered, on an old hypothesis re cultural differences in perspective taking. The introduction presents a very clear overview of relevant literature, explaining key concepts to a wide audience. However, a perspective on the validity of these measures would strengthen the rationale. The addition of the altercentric condition to the director task is a great advantage, but the issue of validity remains. The study could be presented more briefly, but adds to our understanding of cultural similarities.

Main issues

1. What is the ecological validity of the director task? What are associations with other indices (e.g. proxy reports) of social perspective taking? E.g., in clinical groups (like autism) the perspective task has had inconsistent results.

RESPONSE: We have made changes to page 28 to address this point more explicitly. Part of this discussion was included as a footnote as we feel the main text should be focused on the cognitive and cultural narrative.

2. The authors emphasize the focus on the use of perspective but do not provide any account of the empirical literature on cultural differences in this domain.

RESPONSE: We have put forward our interpretation of the Wu & Keysar (2007) and Wu, Barr et al (2013) studies in the introduction (pages 5 and 6). As far as we are aware, these are the only studies investigating cultural difference in perspective-use (rather than ToM inference or reasoning). A related study is Luk et al (2012) which investigated the effect of priming on bilingual/bicultural individuals. We discussed this study in the general discussion section along with discussion about factors affecting perspective-use and egocentrism (pages 27-28), as this study did not directly investigate cultural differences.

3. What is the relative importance of eye movement/RT versus behavioral outcomes, would behavioral outcomes not be more meaningful? Is there a rationale for focusing on eye movement other than methodological (sensitivity?)

RESPONSE: We have added an explanation to page 11. We have kept the explanation brief as inclusion of eye-tracking is standard in the field of referential communication. Eye movement data offer the possibility of capturing effects occurring earlier than the behavioural responses, it also offers the possibility of fine-grained of time course analysis. In this case our preliminary timecourse analysis did not provide additional information to the effects already revealed by the aggregated eye movement measures we report in the manuscript. However, the raw data could be shared with others for further analysis (much like the Wu, Barr et al., 2013 study).

4. Both tasks aim to measure perspective taking, but show very weak correlation. What does this say about their validity? Would it not have been better to include additional measures to provide information on the cultural

RESPONSE: The final sentence was not complete in the review text sent to us. We guess that the reviewer may have intended to refer to cultural identity. We agree that specific measures of cultural identity or even self-construal would have been informative. Unfortunately this did not occur to us when we designed the study.

On the weak/non-sig. correlation between the measures of perspective-taking: we think it doesn't necessarily say much about task validity (as discussed, the tasks have been shown to relate to other measures of social competence and/or functioning), rather it highlights the different sub-processes involved in perspective-taking. The level 1 visual-perspective taking task taps into perspective calculation which occurs relatively quickly. The director task requires the use of such calculated information and the integration of this information with incoming linguistic input. Therefore the weak correlation is perhaps not that surprising. We have clarified this on page 30.

Minor issues.

1. the paper could be shorter, in particular the intro and methods/results. Some methods sections could be moved to appendix

RESPONSE: We have shortened the methods and results sections by moving some of the information to supplementary materials.

2. The info on participants is minimal, what were recruitment procedures, were any analyses ran to justify the level of cultural identification among the participants?

RESPONSE: We have added information about recruitment to page 10. We did not include measures of cultural identity. However, all participants were informed of the cross-cultural nature of the study at debriefing. No participants had declared that they felt unrepresentative of their culture. While this cannot substitute culture identity measures, our understanding is that such practice is not out of line with existing cross-cultural studies.

3. the main effect of culture on error rate is not emphasized in the discussion. How can this contrast be explained given the overall pattern of results?

RESPONSE: For the brevity and narrative of the manuscript we had focused on culture related effects. However, to facilitate interpretation of the effects, we have now included a footnote on page 25.

4. The discussion could be more condensed, several points are repeated, and the studies could be put in a wider theoretical and methodological framework.

RESPONSE: We have now edited some of the existing discussion and included a broader discussion about ToM in independent versus interdependent cultural frameworks on page 31.

Appendix B

Dear Prof Hamilton,

Thank you for accepting our manuscript for publication. We have made the minor revision suggested by Reviewer 1, and highlighted the edit on page 13. We used the wording suggested by Reviewer 1 with the exception that it should read “4000ms after the offset of the sentences” as opposed to “onset”.

Thank you again for considering our manuscript.

Yours Sincerely,

Jessica Wang

Philip Tseng

Chi-Hung Juan

Steven Frisson

Ian Apperly